# Molecular Signatures for Combined Targeted Treatments in Diffuse Malignant Peritoneal Mesothelioma

**DOI:** 10.3390/ijms20225817

**Published:** 2019-11-19

**Authors:** Antonino Belfiore, Adele Busico, Fabio Bozzi, Silvia Brich, Elena Dallera, Elena Conca, Iolanda Capone, Annunziata Gloghini, Chiara C. Volpi, Antonello D. Cabras, Silvana Pilotti, Dario Baratti, Marcello Guaglio, Marcello Deraco, Shigeki Kusamura, Federica Perrone

**Affiliations:** 1Laboratory of Molecular Pathology, Department of Pathology, Fondazione IRCCS Istituto Nazionale dei Tumori, via Venezian 1, 20133 Milan, Italy; antonino.belfiore@istitutotumori.mi.it (A.B.); adele.busico@istitutotumori.mi.it (A.B.); fabio.bozzi@istitutotumori.mi.it (F.B.); silvia.brich@istitutotumori.mi.it (S.B.); elena.dallera@istitutotumori.mi.it (E.D.); elena.conca@istitutotumori.mi.it (E.C.); iolanda.capone@istitutotumori.mi.it (I.C.); annunziata.gloghini@istitutotumori.mi.it (A.G.); chiara.volpi@istitutotumori.mi.it (C.C.V.); silvana.pilotti40@gmail.com (S.P.); 2Department of Pathology, Fondazione IRCCS Istituto Nazionale dei Tumori, via Venezian 1, 20133 Milan, Italy; antonello.cabras@istitutotumori.mi.it; 3Peritoneal Surface Malignancy Unit, Fondazione IRCCS Istituto Nazionale dei Tumori, via Venezian 1, 20133 Milan, Italy; dario.baratti@istitutotumori.mi.it (D.B.); marcello.guaglio@istitutotumori.mi.it (M.G.); marcello.deraco@istitutotumori.mi.it (M.D.); shigeki.kusamura@istitutotumori.mi.it (S.K.)

**Keywords:** diffuse malignant peritoneal mesothelioma, mTOR/PIK3, MET, Axl, EGFR family

## Abstract

Background—There are currently no effective therapies for diffuse malignant peritoneal mesothelioma (DMPM) patients with disease recurrence. In this study, we investigated the biology of DMPM by analyzing the EGFR family, Axl, and MET, in order to assess the presence of cross-talk between these receptors, suggesting the effectiveness of combined targeted treatments in DMPM. Method—We analyzed a series of 22 naïve epithelioid DMPM samples from a single institute, two of which showed higher-grade malignancy (“progressed”). EGFR, HER2, HER3, Axl, and MET activation and expression were investigated by biochemical analysis, real-time PCR immunofluorescence, immunohistochemistry, next-generation sequencing, miRNA, and mRNA in situ hybridization. Results—In most DMPMs, a strong EGFR activation was associated with HER2, HER3, Axl, and MET co-activation, mediated mainly by receptor heterodimerization and autocrine-paracrine loops induced by the expression of their cognate ligands. Axl expression was downregulated by miRNA34a. Mutations in MET Sema domain were exclusively found in two “progressed” DMPMs, and the combined Axl and MET inhibition reduced cellular motility in a DMPM cell line obtained from a “progressed” DMPM. Conclusion—The results indicate that the coordinated activity of multiple cross-talks between RTKs is directly involved in the biology of DMPM, suggesting the combined inhibition of PIK3 and mTOR as an effective strategy that may be easily implemented in clinical practice, and indicating that the combined inhibition of EGFR/HER2 and HER3 and of Axl and MET deserves further investigation.

## 1. Introduction

Diffuse malignant peritoneal mesothelioma (DMPM) is a rare and locally aggressive neoplasm that is traditionally considered to be a rapidly lethal disease. Aggressive cytoreductive surgery (CRS), followed by hyperthermic intraperitoneal chemotherapy (HIPEC), is currently the standard therapy, with a median survival time that ranges from 34 to 92 months [1,2,3]. These results are fairly better than those obtained with palliative systemic chemotherapy (9–13 months) [4,5]; nevertheless, a remarkable number of patients still relapse and ultimately die from this rapidly progressing disease.

The discovery of new targeted therapies could be the key for improving the prognosis of patients affected by DMPM, which, as is already known, is relatively resistant to traditional chemotherapy.

In recent years, studies on DMPM genomic profiling have improved our understanding of the molecular biology of this rare tumor and have identified potential therapeutic targets. In particular, these studies have shown that DMPM is characterized by mutations in BRCA1 associated protein 1 (BAP 1), Neurofibromin 2 (NF2), DEAD-Box Helicase 3 X-Linked (DDX3X), SET Domain Containing 2, Histone Lysine Methyltransferase (SETD2) genes, as well as by the loss of 3p21 locus, which includes chromatin modifiers and epigenetic regulatory genes such as BAP1, SETD2, SWI/SNF-related, matrix-associated, actin-dependent regulator of chromatin subfamily C member (SMARCC1), and Polybromo 1 (PBRM1) [6,7]. Interestingly, DMPMs harboring 3p21 locus or BAP1 loss showed different expressions of a set of genes involved both in chromatin remodeling and DNA damage-repair mechanisms compared to DMPMs with no such mutations, suggesting a potential sensitivity to inhibitors of epigenetic modifiers [6]. Moreover, DMPMs harboring 3p21 locus or BAP1 loss showed an inflammatory tumor microenvironment and expression of programmed cell death 1 (PD1)/programmed cell death 1 ligand (PD-L1), so this subset of DMPMs may be eligible for immune checkpoint blockade therapies.

In addition, anaplastic lymphoma receptor tyrosine kinase (ALK) rearrangements have recently been described in a small subset of younger women affected by DMPM (3%), suggesting that these selected patients may benefit from ALK inhibitors [8].

Finally, a further molecular alteration observed in DMPM is the activation/expression of mammalian target of rapamycin (mTOR) and phosphatidylinositol-4,5-bisphosphate 3-kinase catalytic subunit alpha (PIK3CA) signaling [9], and a gene expression profile study revealed that the overexpression of this pathway correlated with shortened survival in DMPM patients [10]. Correspondingly, the combined inhibition of PI3K and mTOR signaling was effective in two young women with indolent papillary DMPM, allowing long-term survival despite disease recurrence [11].

Our molecular analyses of receptor tyrosine kinases (RTKs) epidermal growth factor receptor (EGFR), platelet-derived growth factor receptor (PDGFR) B and A, and their downstream effectors AKT serine/threonine kinase 1 (AKT), extracellular signal-regulated kinase (ERK), mTOR, ribosomal protein S6, and eukaryotic translation initiation factor 4E binding protein 1 (4EBP1) in DPMP showed a ligand-dependent activation and co-activation of EGFR and PDGFRB, as well as an activation of the downstream mTOR/PIK3CA/AKT/ERK pathways, supporting a combined treatment involving RTK and mTOR inhibitors, rather than the inhibition of just the EGFR pathway, as we confirmed in vitro in a human DMPM cell line [9]. Furthermore, in a new series of surgical epithelioid DMPMs (E-DMPM), we observed c-Met/hepatocyte growth factor receptor (MET) activation, suggesting that MET could cooperate with E-/P-cadherin and MYC Proto-Oncogene, BHLH transcription factor (c-MYC) in mesenchymal-epithelial reverse transition (MErT), a process that confers stemness and plasticity to the cells producing a hybrid phenotype [12].

Starting from these data, in the present study, we aimed to improve our understanding of the role of RTK in DMPM biology. In particular, we extended the analysis to both EGFR family members and RTKs, including Axl and MET, to explore if there is a cross-talk between EGFR and other RTKs that could further support the proposal of combined instead of single-agent treatments in DMPM disease.

## 2. Results

### 2.1. HER Family Analysis

First, we explored the expression/activation of the RTK members of the Erb-B Receptor Tyrosine Kinase (HER) family in 12 of the 22 frozen DMPMs, using the EGFR Phosphorylation Antibody Array. This assay revealed EGFR, HER2, HER3, and HER4 expression in all analyzed cases. The phosphorylation sites detected are as follows: HER2 at residues T686 and S1113 in 12 (100%) cases; HER3 at residue T1289 in 10 (83%) cases; and HER4 at residue T1284 in 9 (75%) cases (Figure 1A). Unexpectedly, EGFR activation (at residues T1173 and T845) was observed in only 3 (25%) cases. 

Based on these results, we decided to further investigate the expression/activation status of the HER family members and the molecular mechanisms underlying receptor activation in the whole series of 22 DMPMs.

#### 2.1.1. EGFR

Consistent with our previous findings [9], IP and WB experiments revealed EGFR phosphorylation in 16 out of 21 (76%) frozen cases and EGFR expression in 20 out of 21 (95%) cases (Figure 1B). Moreover, it was possible to confirm the absence of *EGFR* mutation in 16 FFPE DMPMs by highly sensitive NGS, as we had previously confirmed by direct sequencing, similarly to Karla et al. [13]. NGS revealed also *BAP1* and *NF2* mutations in 25% and 19% of cases, respectively.

We assessed the expression of the main EGFR ligands in 22 cases by real-time PCR and observed transforming growth factor alpha (TGFα) expression in all (100%) cases, amphiregulin in 20 (91%) cases, and heregulin in 14 (64%) cases. 

#### 2.1.2. HER2

Sixteen out of 22 (73%) cases showed HER2 phosphorylation, and HER2 expression was observed in all samples but one (95%) (Figure 1C). HER2/EGFR co-immunoprecipitation was finally observed in 12 out of 22 (54%) cases, providing evidence of protein heterodimerization (Figure 1C).

On FFPE material, HER2 protein was not detectable by IHC and NGS revealed only one mutation p.A386D (6%) in *HER2* exon 10 (case #16). This new mutation is located in the L-receptor domain that shapes the ligand-binding site; however, this mutation is predicted to be functionally benign.

#### 2.1.3. HER3

Seventeen out of 22 (77%) cases showed HER3 phosphorylation, and 21 (95%) showed HER3 expression (Figure 1D). Because HER3 shows a low level of kinase activity and its “on state” is in heterodimers conformation, we investigated HER3/EGFR co-immunoprecipitation. The presence of HER3/EGFR heterodimerization (Figure 1E) was confirmed by EGFR IP: after incubation with anti-HER3 antibody, the expected band appeared on the filter in 16 cases out of 22 (73%).

A similar procedure was performed by using HER2 antibody in the WB experiments, and evidence of HER3/HER2 co-immunoprecipitation was observed in 11 of 19 (58%) cases (Figure 1F).

We also performed IF assay on case #13 frozen tissue, detecting HER3 and EGFR expression at membrane level, as well as HER3 and EGFR co-expression (Figure 1G).

Starting from FFPE material, in all 13 DMPMs (100%) analyzed by IHC, HER3 immunostaining involving both epithelial (membranous staining) and stromal components (Figure 1H) was observed.

The expression of the HER3 ligand heregulin was observed in 10 (45%) cases, and a new damaging p.P30L mutation in *HER3* exon 2 (case #8) was found by NGS.

### 2.2. Phosphorylation Antibody Array 

In addition to EGFR family, we explored the activation of a set of 49 RTKs in 12 frozen DPMPs. A strong EGFR phosphorylation was confirmed in all cases, and the activation of other RTKs was observed, albeit at a lower level of phosphorylation than EGFR. Axl receptor tyrosine kinase (Axl) was found to be active in 11 (92%) cases, receptor-like tyrosine kinase (RYK) in 7 (58%), tyrosine kinase with immunoglobulin-like and EGF-like domains 1 (Tie) in 6 (50%), PDGFRB and HER2 in 5 (42%), macrophage colony stimulating factor receptor (M-CSFR), receptor tyrosine kinase-like orphan receptor 2 (ROR2) and EPH receptor B2 (EphB2) in 3 (25%), EphB3 in 2 (17%), and PDGFRA, vascular endothelial growth factor receptor 2 (VEGFR2), insulin receptor (IR), insulin-like growth factor 1 receptor (IGF1R), EphB6, and erythropoietin-producing hepatocellular carcinoma receptors (EphR) in 1 (8%) case. Unexpectedly, no MET activation was observed, in contrast with our previous data [12].

### 2.3. Axl Analysis

Because of the frequent occurrence of Axl activation, we decided to further analyze this receptor. IP/WB experiments revealed Axl phosphorylation in 18 of 21 cases (86%), as well as Axl expression (Figure 2A). Evidence of Axl/EGFR heterodimerization was observed by co-IP in 17 of 22 (77%) cases (Figure 2B). This finding is consistent with IF experiments performed on the cryopreserved material from case #13, which showed both Axl and EGFR expression and co-expression at the membrane level (Figure 2C).

By ISH and IF assays on FFPE material, we compared Axl expression of two DMPMs (cases #3 and #12) and of the two “progressed” DMPMs (cases #16 and #17) showing a greater morphological grade of malignancy than DMPM. Interestingly, both the “progressed” DMPM cases showed higher RNA expression and, as a consequence, higher Axl protein levels (Figure 2D).

To explore further molecular mechanisms of Axl activation other than heterodimerization, the expression of growth arrest specific 6 (GAS-6), the ligand of Axl, was assessed by real-time PCR in 18 of 19 (95%) cases with no evidence of Axl mutation by NGS analyses.

### 2.4. MET Analysis

Even though the RTK array failed to reveal MET phosphorylation, we previously observed MET activation in E-DMPM [12]. We therefore performed MET WB analysis and found activation and expression in 11 of 21 (52%) and in 15 of 21 (71%) cases, respectively (Figure 2E). Evidence of EGFR/MET heterodimerization was observed by co-IP in 15 of 21 (71%) cases (Figure 2F). 

On FFPE material, MET expression was confirmed by IHC in all cases but one (15/16 = 94%). In particular, “progressed” DMPMs showed lower MET protein levels than DMPMs (Figure 2G). Furthermore, expression of hepatocyte growth factor (HGF), the ligand of MET, was observed in 18 of 19 (95%) cases.

Interestingly, by NGS MET mutations were exclusively found in the two “progressed” DMPMs (p.R359Q case #16 and p.E168D case #17). Even if these *MET* mutations have conflicting interpretations of pathogenicity, they are located in the MET Sema domain that is necessary for HGF binding and receptor dimerization/activation, and mutations in this domain can promote an invasive malignant phenotype [14].

### 2.5. Cell Motility and Proliferation Assay

A SSL1 cell line obtained from fresh material from the “progressed” DMPM case #16 carrying a mutation in MET Sema domain was available [12].

To investigate whether the combined inhibition of Axl and MET can affect DMPM motility and migration, SSL1 confluent cells were treated with BMS-777607, a double MET, and AXL inhibitor. SSL1 cells with no treatment (NT) or treated with DMSO were used as controls. After 24 h, only few SSL1 cells treated with 2 µM and 5 µM BMS repopulated the scratch area, indicating a sensible reduction in cell migration capacity compared with NT- and DMSO-treated cells that completely filled the empty surface left by the scratch (Figure 3).

By contrast, cellular proliferation experiments on SSL1 treated with BMS-777607 showed a lower proliferation than NT only at high dosages (5 and 10 µM) (Figure 3C).

### 2.6. miRNA 34a Analysis

We decided to further investigate a specific miRNA family, i.e., miRNA 34a, by ISH analysis because we found that modulation of MET and Axl expression levels occur through miRNA 34a, as already described in literature [15,16]. In our samples, we could confirm a correlation between the miRNA 34a signal levels and MET expression. In particular, in two DMPMs (cases #4 and #8) miRNA 34a and MET expressions were found to be similar, whereas in “progressed” DMPMs (cases #16 and #17) both miRNA 34a and MET expressions were significantly reduced (Figure 2G). On the other hand, we observed an inverse correlation between Axl (Figure 2D) and miRNA 34a (Figure 2G) levels. In particular, there was no Axl expression in DMPM #3 that showed high levels of miRNA 34a, whereas high Axl expression coupled with absence of miRNA 34a signals was observed in the two “progressed” DMPMs. 

## 3. Discussion 

In this study, we found evidence of co-expression and co-activation of different RTKs in DMPM, and these findings encourage the use of combined rather than single inhibitor agents.

On DMPM surgical specimens, strong EGFR activation was often coupled with HER2 and HER3 activation, mediated by receptor heterodimerization and by an autocrine–paracrine loop mediated by the expression of the cognate ligands. The molecular mechanisms we investigated in this study could explain the low efficacy of single-agent anti-EGFR therapy reported in DMPM patients [17,18], despite predominant EGFR overexpression/activation. Our data support the investigation of EGFR pan inhibitors or combined treatments. It is worth mentioning that pan EGFR inhibitors available in clinical practice, such as afatinib, dacomitinib, or poziotinib, target EGFR, HER2, and HER4 but not HER3, which may indeed contribute to mediating DMPM cell survival signaling. In particular, the expression of the HER3 ligand heregulin that we observed in our DMPM series may be able to induce resistance to EGFR-inhibitors, as has been observed in lung cancer [19]. This heregulin-mediated resistance could be overcome by the combination of EGFR-inhibitor and the anti-HER3 monoclonal antibody patritumab that potently inhibits cell proliferation, according to in vitro and in vivo experimental data [19].

In addition to HER family, we also observed co-activation of other RTKs, such as Axl, PDGFRB, RYK, Tie, and ROR2. Among them, we focused on Axl because its phosphorylation was observed in all analyzed DMPMs, and it is known that this receptor may cause resistance to EGFR-targeted therapy [20,21,22], e.g., by regulating the nuclear translocation of EGFR [23]. IP-WB, ISH, and IF experiments confirmed frequent expression and activation of Axl in our DMPM series. We also found that Axl activation is related to both heterodimerization of Axl-EGFR and the expression of the ligand GAS-6.

Furthermore, we confirmed frequent MET expression/activation mediated by MET-EGFR heterodimerization, expression of the cognate ligand HGF in most of DMPM, as well as, exclusively in “progressed” DMPMs, MET mutations on the Sema domain. Overall, our findings are in agreement with evidence obtained in pleural mesothelioma cell lines that displayed co-activation of EGFR, Axl, and MET, supporting the hypothesis of a higher responsiveness to combined targeted therapies [24,25]. Our results on the SSL1 cell line, obtained starting from the fresh material of #16 “progressed” DMPM, showed that the inhibition of both Axl and MET decreased cell proliferation only at high drug concentration. However, even at low concentration, this drug significantly reduced SSL1 motility, in agreement with the widely accepted notion that both Axl and MET promote cellular migration and invasion [26] and with the finding that mutation in MET Sema domain favors an invasive malignant/aggressive phenotype [14].

As a miRNA 34a-mediated down regulation of both Axl and MET has been previously reported in DMPM cell lines [15,16], we explored the expression of miRNA 34a on surgical samples, using ISH. We confirmed an inverse correlation between miRNA 34a and Axl expression, since high miRNA 34a levels were coupled with absence of Axl expression and vice versa; by contrast, we observed a positive correlation between miRNA 34a and MET expression. Interestingly, in the two “progressed” DMPMs, we detected lower MET/miRNA 34a and higher Axl expression levels than in DMPM. This molecular signature of “progressed” DMPM may favor epithelial mesenchymal transition (EMT) together with the previously described decreased E-cadherin and increased MYC expression [12]. It has indeed been reported that miRNA 34a inhibits EMT [27], whereas Axl favors EMT inducing the expression of stemness markers [28,29], which in turn generate cells with stem cells properties [30]. Both stemness and EMT are implicated in cancer resistance to treatment [31,32]. Accordingly, Axl expression is responsible for resistance to chemotherapy [29], while anti-Axl monoclonal antibodies enhance the antitumor effect of chemotherapy, in addition to reversing resistance to EGFR-inhibitors [20,21]. Thus, we can speculate that, at least, the “progressed” DMPMs showing MET mutations and Axl overexpression/activation may benefit from chemotherapy combined with anti Axl/MET treatment. 

On the other hand, the activation of multiple RTKs characterizing DMPM led to consider the inhibition of downstream RTK common effectors as an effective treatment. On the basis of the EGFR, PDGFRA, PDGFRB, mTOR, PIK3CA and AKT activation we previously observed in DMPM specimens, we had previously identified the antiproliferative effect of the mTOR inhibitor RAD001 in a human DMPM cell line [9]. The activation of the EGFR family, Axl and MET, extensively detailed in this study, further supports the molecular rationale underlying the efficacy of mTOR inhibition. However, results of a phase II study in advanced malignant pleural mesothelioma were disappointing as the single agent everolimus had limited clinical activity [33], probably because of the activation of AKT negative-feedback after mTOR inhibition, as demonstrated in vitro [24]. Consistent results had been obtained on both DMPM and pleural mesothelioma cell lines where the antiproliferative effect of combined mTOR and PI3K inhibition was stronger than that induced by single agents [10,34]. Notably, the safe, modest but durable antitumor activity of the dual inhibitor of PI3K and mTOR apitolisib has already been confirmed in patients with advanced solid tumor including mesothelioma [35]. This therapeutic strategy has been shown to be effective in two young women with papillary indolent DMPM enabling long-term survival despite disease relapse [11].

## 4. Material and Method

### 4.1. Patients and Samples

Our institution, the Fondazione IRCCS Istituto Nazionale dei Tumori di Milano, is an international reference center for DMPM, ensuring both an adequate recruitment of patients affected by this rare malignancy and uniformity of treatment protocol. We analyzed naïve DMPM specimens taken from 22 patients treated with CRS and HIPEC between 2011 and 2014. Five patients were female and 17 were male; ages ranged from 47 to 77 years (median age 64 years). This study was performed according to the clinical standards of the 1975 and 1983 Declaration of Helsinki and was approved by the Ethical Committee of Fondazione IRCCS Istituto Nazionale dei Tumori (INT), Milan, Italy (n DMPMtarget012, 02/04/2012). According to our standard practice, all patients had signed an informed consent to the use of their data for research purposes. 

Diagnoses were based on morphological and immunophenotypic criteria including: (1) immunoreactivity for calretinin (Ventana Medical Systems, Tucson, AZ, USA, prediluted), keratin 5 and 6 (clone D5/I6B4, Zymed Laboratories, San Francisco, CA, USA), podoplanin (11-009PE AngioBio, San Diego, CA, USA), and Wilms’ tumor 1 proteins (C-19, Santa Cruz Biotechnology, Dallas, TX, USA); (2) claudin 4 null immunophenotype (clone 3E2C1, Zymed Laboratories, San Francisco, CA, USA). All cases presented epithelioid features; #16 and #17 were classified as “progressed” because they displayed higher malignancy than DMPM [12].

### 4.2. Biochemical Analysis

Phosphorylation of EGFR and 49 RTK was assessed by EGFR Phosphorylation Antibody Array I (RayBiotech, Peachtree Corner, GA, USA) and Phosphorylation RTK array (R&D Systems, Minneapolis, MN, USA), respectively. In accordance with the manufacturer’s instructions, we used 1 mg of protein lysate extracted from frozen material, as previously described [9]. EGFR, HER2, and HER3 immunoprecipitation (IP) and Western blotting (WB) experiments were performed as previously described by Cortelazzi et al. Axl anti-Rabbit monoclonal antibody (clone C44G1, Cell Signaling Technology, Danvers, MA, USA) was used for the WB, anti-phosphotyrosine mouse monoclonal antibody (clone 4G10, 1:3000, Merck Millipore, Burlington, MA, USA) was used to test Axl phosphorylation, and Axl anti-Rabbit monoclonal antibody (C44G1, 1:1000, Cell Signaling Technology, Danvers, MA, USA) was used to test Axl expression. To detect MET activation and expression, WB experiments were performed, using P-MET anti-Rabbit polyclonal antibody (phospho Y1230 + Y1234 + Y1235, antibody ab 5662, 1:1000, Abcam, Cambridge, UK) and MET anti-Rabbit monoclonal antibody (SP44, 1:1000, Spring Bioscience Roche, Zurich, CH).

### 4.3. Real-Time PCR 

After RNA extraction and reverse transcription from frozen samples, the cDNA of EGFR ligands, HER3, Axl, MET, and the housekeeping β2-microglobulin were detected by real-time PCR, using a TaqMan assay (ABIPRISM 5700 PCR Sequence Detection Systems, Thermo Fisher Scientific, Waltham, MA, USA). 

### 4.4. Immunofluorescence (IF)

EGFR/Axl and EGFR/HER3 co-expression was investigated by IF. Two 4 µm sections of frozen surgical samples were incubated with polyclonal anti-human Axl antibody AF154 (R&D Systems, Minneapolis, MN, USA) at 1:200 dilution, monoclonal anti-human EGFR Pharma DX Dako (Agilent Technologies, Santa Clara, CA, USA), and rabbit polyclonal anti-human c-erbB3 20-786-255900 (Aviva System Bio Corp, San Diego, CA, USA) at 1:50 dilution. Tissue sections were incubated with the specific secondary Alexa Fluor antibodies (Alexa Fluor 488 and Alexa Fluor 546, Thermo Fischer Scientific, MA, USA), at room temperature, for one hour, and then they were mounted, using Vectashield mounting medium with DAPI (Vector Labs, Burlingame, CA, USA). The slides were then incubated overnight at −20 °C. Samples were examined with a Leica DM6000B microscope equipped with a 100 W mercury lamp, excitation filters length Spectrum Orange (546 nm), Spectrum Green (488 nm), and DAPI. The images were acquired at different magnifications (20× and 40×) oil immersion and analyzed, using Cytovision software. Images were sequentially collected in a single channel, in order to reduce fluorescence cross-talk.

### 4.5. Immunohistochemistry (IHC)

IHC was carried out on FFPE 2 µm sections, using an automated immunostainer (BenchMark Ultra, Ventana Medical Systems, Inc, Tucson, AZ, USA), in accordance with the manufacturer’s instructions. We used the polyclonal rabbit anti-human erbB2 oncoprotein antibody (clone A0485, Agilent Dako Santa Clara, CA, USA) at 1:1000 dilution, the c-erbB3 polyclonal rabbit antibody (clone 20-786-255900, GenWay Biotech, San Diego, CA, USA) at 1:50 dilution, and the MET anti-Rabbit monoclonal antibody (clone SP44, 1:50, Spring Bioscience Roche, Zurich, CH).

### 4.6. Next-Generation Sequencing (NGS)

Genomic DNA was extracted from both FFPE tumor and nontumoral tissue surrounding the DMPM, using the GeneRead DNA FFPE kit (Qiagen, Hilden, Germany, http://www.qiagen.com Cat. n. 180134).

Mutations in genes encoding for RTKs and their downstream effectors were assessed by the Ion AmpliSeq Comprehensive Cancer Panel v2 (Thermo Fisher); this panel is designed to target 409 tumor-suppressor genes and oncogenes exons frequently cited and frequently mutated. 

The library was prepared by using IonAmpliSeq Library kit 2.0 (Thermo Fisher), according to the manufacturer’s instructions. Sequencing was performed on the ION S5 XL System (Thermo Fisher), using Ion 540 Chipsand Ion 540™ Kit-Chef, according to the manufacturer’s instructions (MAN0010846). Data were processed, using Torrent Suite™; the variant calling from sequencing data was generated by using the Variant Caller plugin. Resulting variants were annotated with the Ensemble Variant Effect Predictor pipeline, Ion Reporter™ analysis software, COSMIC database, dbSNP database, and ClinVar db of the National Center for Biotechnology Information (http://www.ncbi.nlm.nih.gov/clinvar/). The filtered variants were examined using the Integrative Genomic Viewer IGV tool [36]. 

### 4.7. miRNA in Situ Hybridization (ISH)

miRNA studies were performed as previously described [37]. Briefly, after tissue preparation and permeabilization, tumor sections were hybridized with double-DIG-LNA probes for miR-34a (cat# 38487-15, Exiqon, Vedbaek, Denmark), according to manufacturer’s instructions. Signal detection was performed by using the OptiviewDAB Detection Kit (Ventana Medical Systems, Tucson, AZ, USA) suitable on Ventana BenchMark ULTRA (Ventana Medical Systems).

### 4.8. mRNA in Situ Hybridization (ISH)

ISH for AXL mRNA was performed by hand, using the RNAscope HPV kit (Advanced Cell Diagnostics, Inc., Hayward, CA), according to the manufacturer’s instructions. In brief, FFPE 5 µm sections were pretreated with heat and protease before hybridization with the AXL probe (cat#602131, Advanced Cell Diagnostics, Minneapolis, MN, USA). A horseradish peroxidase-based signal amplification system was used to bind the target probes; signal detection development was performed with diaminobenzidine. Positive staining was identified as brown, with multiple dots localized in the nucleus and/or cytoplasm. Control probes for the bacterial gene 4-hydroxy-tetrahydrodipicolinate reductase (DapB), negative control, and for the housekeeping gene ubiquitin C (UbC), positive control for evidence of preserved RNA, were also included on each run.

### 4.9. In Vitro Analysis

The SSL1 spontaneous DMPM cell line previously obtained from a fresh sample of “progressed” DMPM #16 was used [12].

The double AXL and MET inhibitor BMS-777607 (CAT: S1561, SELLEK) was prediluted to 5 mM in DMSO. To measure cellular proliferation, cells were treated with 1, 5, and 10 µM BMS-777607. After 5 days of treatment, the cells were detached, using a 1% trypsin-EDTA solution (Cat. No. 15400, Thermo Fischer), and counted, using a standard trypan blue assay. 

Cell motility was assessed by a standard scratch assay. Four different SSL1 cell cultures were grown to confluence, and a wound was created into each dish by using sterile pipette tips to draw a straight line. Nonadherent cells were rinsed away, using PBS. Adhered cells were treated with 2 and 5 µM BMS-777607. As control, two culture dishes with no treatment (NT) and DMSO, respectively, were used. Wound-closing was observed under light microscope. Cells were photographed at CM (t = 0 h) and at 24 h (t = 24 h) in the same location.

## 5. Conclusions

In conclusion, our findings indicate that the coordinated activity of multiple cross-talking RTKs is directly involved in the biology of DMPM. This evidence strongly suggests that the combined inhibition of either EGFR/HER2 and HER3 or Axl and MET or PIK3 and mTOR might be an effective therapeutic strategy, worth of further clinical investigation.

## Figures and Tables

**Figure 1 ijms-20-05817-f001:**
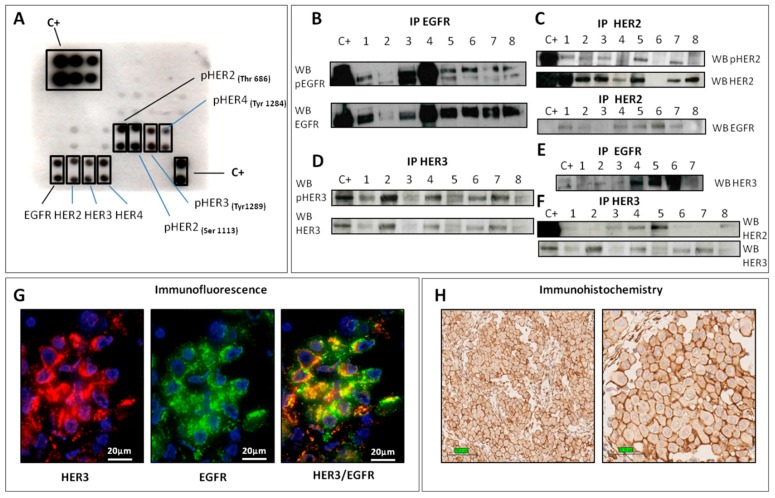
HER family analysis. (**A**) EGFR phosphorylation antibody array revealed phosphorylation of HER2 (T686 and S1113), HER3 (T1289), and HER4 (T1284), as well expression of EGFR, HER2, HER3, and HER4. One milligram of total protein lysate was used for IP experiments, and 20 µl of immunoprecipitated protein was used for WB. A431 (EGFR/HER3) and SKBr3 (HER2) cell lysates were used as positive control for p- and total protein. (**B**) EGFR IP/WB experiments confirmed phosphorylation (upper box) and expression (bottom box) of EGFR. (**C**) HER2 IP/WB experiments confirmed phosphorylation (upper box) and expression (middle box) of HER2; HER2/EGFR co-IP experiments revealed HER2-EGFR heterodimerization (bottom box). (**D**) HER3 IP/WB experiments confirmed phosphorylation (upper box) and expression (bottom box) of HER3. (**E**) EGFR/HER3 co-IP experiments revealed HER3-EGFR heterodimerization. (**F**) HER3/HER2 co-IP experiments revealed HER3-HER2 heterodimerization; HER3 WB revealed HER3 expression. (**G**) HER3 (red) and EGFR (green) IF provided evidence of HER3 and EGFR co-expression (yellow). (**H**) IHC showed HER3 immunoreactivity. Legend: IP: immunoprecipitation; WB: Western blotting; IF: immunofluorescence; IHC: immunohistochemistry.

**Figure 2 ijms-20-05817-f002:**
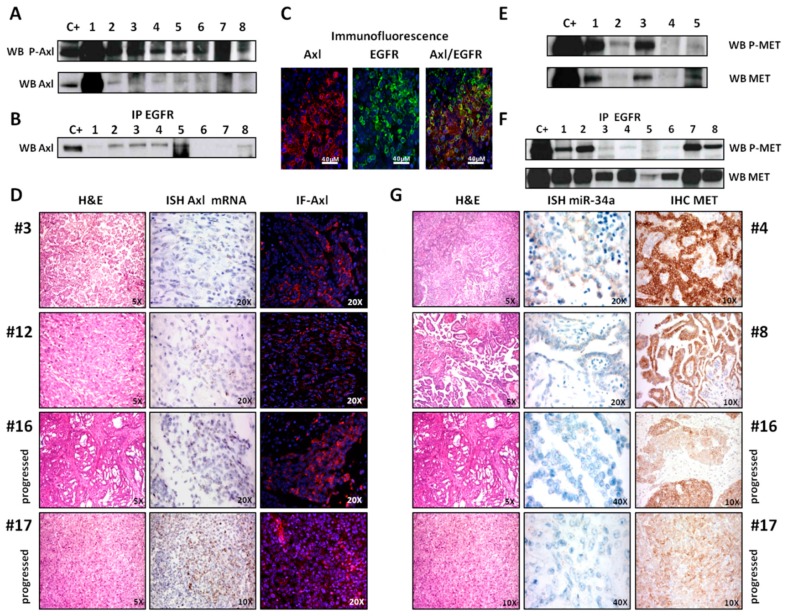
Axl, MET, and miR-34a analysis. (**A**) WB experiments confirmed phosphorylation (upper box) and expression (bottom box) of Axl. (**B**) Axl/EGFR co-IP experiments revealed Axl-EGFR heterodimerization. (**C**) Axl (red) and EGFR (green) IF provided evidence of Axl and EGFR co-expression (yellow). (**D**) Axl mRNA ISH (middle column) and Axl IF (right column) assays showed both higher RNA and protein expression in two DMPMs defined as “progressed” (cases #16 and #17) than two DMPMs (cases #3 and #12). (**E**) MET IP/WB experiments confirmed phosphorylation (upper box) and expression (bottom box) of MET. (**F**) MET/EGFRco-IP experiments revealed MET-EGFR heterodimerization. (**G**) miRNA 34a ISH (middle column) and MET IHC (right column) assays showed both lower miRNA 34a signal levels and MET protein expression in two DMPMs defined as “progressed” (cases #16 and #17) than two DMPMs (cases #4 and #8). Legend: IP: immunoprecipitation; WB: Western blotting; IF: immunofluorescence; ISH: *in situ* hybridization; IHC: immunohistochemistry.

**Figure 3 ijms-20-05817-f003:**
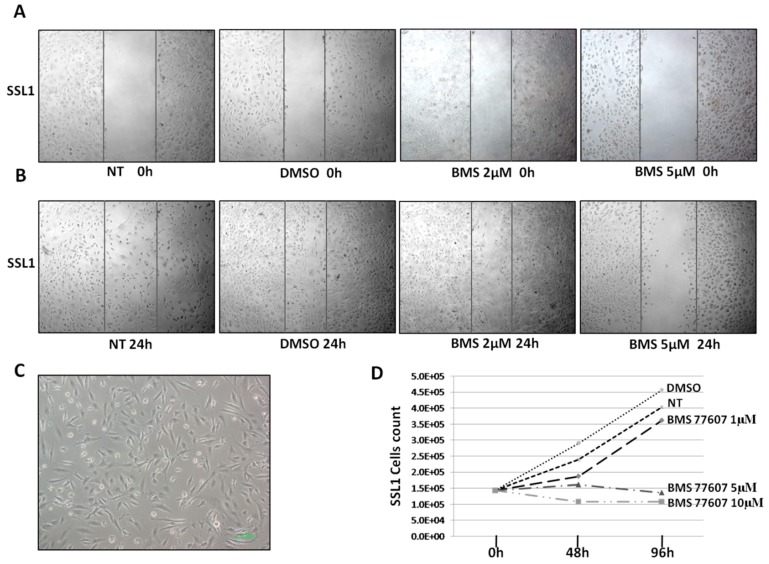
Standard scratch assay: SSL1 cells were grown to confluence, scratched using a sterile tip, and then treated with 2 or 5 µM BMS. Cells with no treatment (NT) or treated with DMSO were used as controls. (**A**) Cells were photographed at t = 0 h and (**B**) after 24 hours of treatment (t = 24 h) in the same location. BMS reduced cell migration. (**C**,**D**) Proliferation assay: SSL1 cells were treated with 1 µM, 5 µM or 10 µM BMS. Cells showed a lower proliferation at 5 μM and 10 μM of BMS. NT cells and DMSO were used as control.

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
