# Peer review of "Molecular Signatures for Combined Targeted Treatments in Diffuse Malignant Peritoneal Mesothelioma"

_ijms, 2019, doi:10.3390/ijms20225817_

Round 1

Reviewer 1 Report

Authors investigated the DMPM patients using IHC.

Data is clear.

In Fig.1 IHC, need more high magnification and show arrow.

Author Response

Response to the reviewer comments

Editor International Journal of Molecular Sciences 

Manuscript  ID ijms-610444 

Dear Editor,

Thank you for facilitating the review of our manuscript entitled “Molecular signatures for combined targeted treatments in diffuse malignant peritoneal mesothelioma”  of  Belfiore et al. We  have made specific modifications (in bold)  to the manuscript based on the reviewer recommendations. Moreover, we read the whole text and corrected some inaccuracies (in yellow).

Please find below  the responses to reviewer comments.

Best regards

Dott. Federica Perrone

Reviewer 1

As suggested, we improved the magnification of the Fig 1 and we showed the scale bar. 

Reviewer 2 Report

Belfiore Antonino et al here did a comprehensive analysis of activation status of EGFR family, Axl and MET in the diffuse malignant peritoneal mesothelioma (DMPM) patients and revealed crosstalk between these RTK signaling.

Many of gene or proteins do not provide a full name when described first in the text. Fig1B-F should be better interpreted. Since IP was done before WB, the input data should be provided here. p-protein and total protein ratio should be quantitated as an indicator of receptor activation. In addition, these growth factor receptors such as EGFR activation is associated with phosphorylation and receptor endocytosis mediated degradation and recycling. Receptor activation needs to be better justified.  Both p-and total protein WB should be done in a total cell lysate, and there are a couple of commercially available good abs. Furthermore, their data quality is bad and some of the interpretation is wrong, eg, Fig1C, lane 6, when IP with HER2, and there is undetectable HER2, but there are highest amount of EGFR is present, that means their abs are bad, they are just nonspecific band. Fig1F, HER3 WB should be included. Same as Fig1E. Fig1G high magnification should be provided and the scale bar is missing. The same applies to other IF and IHC images. In the abstract and methods parts, the author mentioned they used the real-time PCR. However, no relevant result can be found in the text Fig2 and real-time PCR analysis is very important data to show here. Cell proliferation assay and cytotoxicity assay should be included in Fig3. 

Author Response

Response to the reviewer comments

Editor International Journal of Molecular Sciences 

Manuscript  ID ijms-610444 

Dear Editor,

Thank you for facilitating the review of our manuscript entitled “Molecular signatures for combined targeted treatments in diffuse malignant peritoneal mesothelioma”  of  Belfiore et al. We  have made specific modifications (in bold)  to the manuscript based on the reviewer recommendations. Moreover, we read the whole text and corrected some inaccuracies (in yellow).

Please find below  the responses to reviewer comments.

Best regards

Dott. Federica Perrone

Reviewer 2

As suggested, we provided the full name of the genes. In the legend of Fig. 1B-F we specified the quantity of protein we used for immunoprecipitation (1mg) and for WB (20 µl of protein immunoprecipitated). The main goal of our IP/WB experiments was to obtain a “qualitative” evidence of receptor phosphorylation and heterodimerization on DPMP fixed samples. To  achieve this aim, we used the same commercial specific antibodies we had previosuly tested in other series of solid tumors, as documented by the papers published by our laboratory group. As positive control we always used a cell lysate and our results are convincing. Indeed, the result shown in Fig 1C, lane 6, documents the presence of EGFR/HER2 hetodimers composed mainly by EGFR (detectable by WB specific for  EGFR after IP specific for HER2) and only minimally by HER2 (Undetectable by WB specific for  HER2 after IP specific for HER2). If the HER2 antibody had not been specific, it would not have immunoprecipitated HER2 and therefore we would not have obtained any specific signal in EGFR WB. We added HER3 WB in Fig 1F and we modified the relative legend. Regarding the Fig. 1E, we did not even perform EGFR WB, because IP EGFR/WB EGFR experiments showed expression of EGFR in all but one samples. In all figures we provided magnification and the scale bar. We used real- time PCR to exclusively assess the expression of the ligands of EGFR, HER3, Axl and MET. The ligand expression is an important data which suggests the presence of an autocrine-paracrine loop contributing to receptor activation and in the results section we specified the percentage of cases ex pressing a specifi ligand. However, since our real time PCR analysis was qualitative and not quantitative, we think that showing an image of these data would only be an unclear bundle of sigmoid curves where the individual samples are not distinguishable. We showed cell proliferation assay in Fig. 3 and modified the legend of the same Figure. We did not perfom any cytotoxicity assay because we were interested in studying the effect of the drug on cell motility. To this, we used not cytotossic concentrations of the drug.

Round 2

Reviewer 2 Report

The authors' response did not fully address most of the concerns. For example,

 Fig 1C, lane 6 explanation is not acceptable. The alternative could be HER2 ab detected a non-specific protein that binds to EGFR. 

Both p-and total protein WB should be done in a total cell lysate, and there are a couple of commercially available good abs. This was not addressed.

Author Response

Reviewer 2

Fig 1C. We used in our HER2 IP experiments a specific anti-HER2 antibody which was obtained at our institution. It was employed in several experiments reported in peer reviewed manuscripts already published and available in PubMed such as those here after reported:

Antibody-induced activation of p185HER2 in the human lung adenocarcinoma cell line Calu-3 requires bivalency. Srinivas U, Tagliabue E, Campiglio M, Ménard S, Colnaghi MI. Cancer Immunol Immunother. 1993 Jun;36(6):397-402. PMID: 8098992 DOI: 10.1007/bf01742256

p185 HER2/neu epitope mapping with murine monoclonal antibodies. Centis F, Tagliabue E, Uppugunduri S, Pellegrini R, Martignone S, Mastroianni A, Ménard S, Colnaghi MI. Hybridoma. 1992 Jun;11(3):267-76. PMID: 1379973 DOI: 10.1089/hyb.1992.11.267

Selection of monoclonal antibodies which induce internalization and phosphorylation of p185HER2 and growth inhibition of cells with HER2/NEU gene amplification. Tagliabue E, Centis F, Campiglio M, Mastroianni A, Martignone S, Pellegrini R, Casalini P, Lanzi C, Ménard S, Colnaghi MI. Int J Cancer. 1991 Apr 1;47(6):933-7. PMID: 1672668 DOI: 10.1002/ijc.2910470625

We used in all our IP and WB experiments A431 (EGFR/HER3) and SKBr3 (HER2) total cell lysates as positive control for p- and total protein. We added this additional information in the figure 1 legend.